# Form vision from melanopsin in humans

Annette E. Allen [1], Franck P. Martial[1] & Robert J. Lucas[1]

Detection and discrimination of spatial patterns is thought to originate with photoreception by rods and cones. Here, we investigated whether the inner-retinal photoreceptor melanopsin could represent a third origin for form vision. We developed a 4-primary visual display capable of presenting patterns differing in contrast for melanopsin vs cones, and generated spectrally distinct stimuli that were indistinguishable for cones (metamers) but presented contrast for melanopsin. Healthy observers could detect sinusoidal gratings formed by these metamers when presented in the peripheral retina at low spatial ($\leq$0.8 cpd) and temporal ($\leq$0.45 Hz) frequencies, and Michelson contrasts $\geq$14% for melanopsin. Metameric gratings became invisible at lower light levels (<$10^{13}$ melanopsin photons $cm^{-2} sr^{-1} s^{-1}$) when rods are more active. The addition of metameric increases in melanopsin contrast altered appearance of greyscale representations of coarse gratings and a range of everyday images. These data identify melanopsin as a new potential origin for aspects of spatial vision in humans.

---

[1] Division of Neuroscience and Experimental Psychology, School of Biology, Faculty of Biology Medicine and Health, University of Manchester, Manchester M13 9PT, UK. Correspondence and requests for materials should be addressed to A.E.A. (email: Annette.allen@manchester.ac.uk) or to R.J.L. (email: Robert.lucas@manchester.ac.uk)

Visual perception originates with light detection by retinal photoreceptors. Rods and cones in the outer retina dominate light detection, but their activity is augmented by a small number of retinal ganglion cells, which are also directly photosensitive, thanks to expression of the photopigment melanopsin (mRGCs; refs. [1–3]). The best-established function of mRGCs is to provide a signal of ambient light to drive a number of reflex light responses, including circadian photoentrainment and pupil constriction[4–7]. However, the population of mRGCs is anatomically and functionally diverse[8], and a subset of them innervate the primary visual thalamus (dorsal lateral geniculate nucleus; dLGN) in both rodents and primates[9–11]. Accordingly, melanopsin-driven neurophysiological responses to light have also been recorded in both the dLGN and primary visual cortex[9,12–14].

The appearance of melanopsin signals in the primary visual projection raises the question of what contribution(s) melanopsin makes to perceptual vision. A body of data suggest that the signal of ambient light provided by melanopsin is used to adjust aspects of conventional cone-based vision[15–21]. However, experiments in both mice and humans indicate that melanopsin can also influence perception more directly, by providing a sense of 'brightness'[12,14,22] (and perhaps in other ways[23]). For technical reasons, melanopsin-driven perception has, thus far, been studied only with diffuse and/or large featureless stimuli. An important open question therefore is whether melanopsin can contribute to discriminating patterns within a scene (provide form vision) or merely provide an overall impression of scene brightness. Poor temporal resolution is a cardinal feature of melanopsin phototransduction across species[3,10], and this would appear to place a fundamental limit on its utility for spatial discrimination in the face of ongoing changes in direction of view. Nevertheless, direct assessments of the spatiotemporal resolution of melanopsin photoreception under physiological, light adapted, conditions have been notably absent from the literature. We recently addressed that deficit by recording electrophysiological responses in the mouse dLGN to melanopsin-directed spatial patterns[24]. Those recordings revealed that melanopsin has the capacity to encode coarse (low spatiotemporal frequency) patterns. However, direct experimental demonstration of a melanopsin contribution to pattern perception in any species is currently lacking.

Here, we set out to ask whether melanopsin contributes to spatial pattern detection in healthy human subjects. We approached this by using the principles of receptor silent substitution and metamerism to generate carefully calibrated visual stimuli in which spatial contrast for melanopsin can be controlled independent of that for cones. This strategy has been effectively applied to describe the contribution of melanopsin to non-image forming visual responses[25–28] and to reveal melanopsin's role in perception of scene brightness[12,22], but until now has not been applied to an apparatus capable of presenting spatial patterns. To overcome this, we have extended the three primary (red, green, blue; RGB) architecture of conventional visual displays to render images in four spectrally distinct subpixels (primaries; Fig. 1a). The additional degree of freedom allowed by inclusion of a 4th primary allows us to produce patterns that are indistinguishable for cones ('metameric') but contain significant contrast for melanopsin. We show that such patterns are detectable when presented at low spatiotemporal frequencies and high radiance, and can augment the appearance of greyscale images when superimposed upon them. These data support the hypothesis that melanopsin can indeed provide form vision, allowing detection and discrimination of patterns at low spatiotemporal frequencies and influencing the appearance of everyday images.

## Results

**Spatial patterns indistinguishable to cones.** A conceptually straightforward test of the hypothesis that melanopsin contributes to form vision is to ask whether people can detect patterns that are invisible to cones (metameric) but contain contrast for melanopsin. To achieve this, we constructed a display with four primaries (violet, cyan, green and red) allowing independent control over spatiotemporal contrast for melanopsin vs the three classes of cone photoreceptor (long, medium and short wavelength sensitive (LMS) cone). We then employed the published spectral sensitivity of L, M and S cones in a theoretical 'standard observer'[29] to calculate settings for the four primaries that would produce two spectrally distinct outputs that had equivalent effective radiance for each of the cones but differed in 'melanopic' radiance (metameric pairs). We reasoned that patterns formed by mixing these two spectra in different ratios across a projected image should be visible to melanopsin but not cones. However, such an approach does not allow for inter-individual variations in cone spectral sensitivity and/or pre-receptoral filtering that could render such theoretically metameric stimuli visible to cones. We therefore set out to empirically define metameric pairs for each subject. Starting from the calculated metameric pair for the standard observer (target CIE 1931 xy chromaticity: (0.31, 0.33); 214 cd m$^{-2}$), we generated 170 variants of these pairs by making small alterations to the spectral composition of one the constitutive spectra (Fig. 1b). To identify which (if any) of these pairs was truly metameric for any given observer, we used them to generate 170 stationary sinusoidal gratings, at a spatial frequency well within the sensitivity range of cones (3.2 cycles per degree; cpd). These gratings were presented in a random order at one of four orientations in the left peripheral visual field (15° diameter disc viewed monocularly, located 19° temporally and 6.5° superior to a fixation point; Fig. 1c; supplementary fig. 1), and participants were asked to report grating orientation in a 4-alternative forced choice paradigm (summarised in supplementary fig. 1c). For all subjects, at least one spectral pair was sufficiently hard to detect that they incorrectly identified the orientation in 3/3 separate presentations (representative participant shown in Fig. 1e). Notably, for all participants, subtle changes to the starting spectrum were required to achieve metamerism (Fig. 1f). This implies slight variations in effective cone spectral sensitivity in our subjects from that of the theoretical 'standard observer', and confirms the need for individual-by-individual calibration of metameric pairs. Substantial melanopsin contrast was retained in each of the validated metameric pairs (mean ± SD Michelson contrast: 19.0 ± 2.28%).

Metamerism can fail in cones falling in the shadow of blood vessels ('penumbral cones') even when established across the remainder of the retina[30]. One way in which this failure can be apparent is in appearance of a Purkinje-tree image of retinal vasculature. To minimise the possibility of such inadvertent cone contrast, our starting spectral power distributions were designed to minimise penumbral cone contrast (<2% Michelson contrast for L/M/S penumbral cones). Accordingly, no subjects reported appearance of the Purkinje tree. This was not because of a fundamental inability to do so with such stimuli, as retinal vasculature became apparent when gratings were adjusted to elevate penumbral cone contrast (~4.5% penumbral L contrast; Fig. 1g).

**Spatial patterns detected by melanopsin.** Having validated metameric patterns for each subject, we turned to the question of whether they might be visible to melanopsin. Given the dendritic field size of human mRGCs (1.5–3°[31]), and data from mice[24], we expect melanopsin to have much lower spatial acuity than cones.

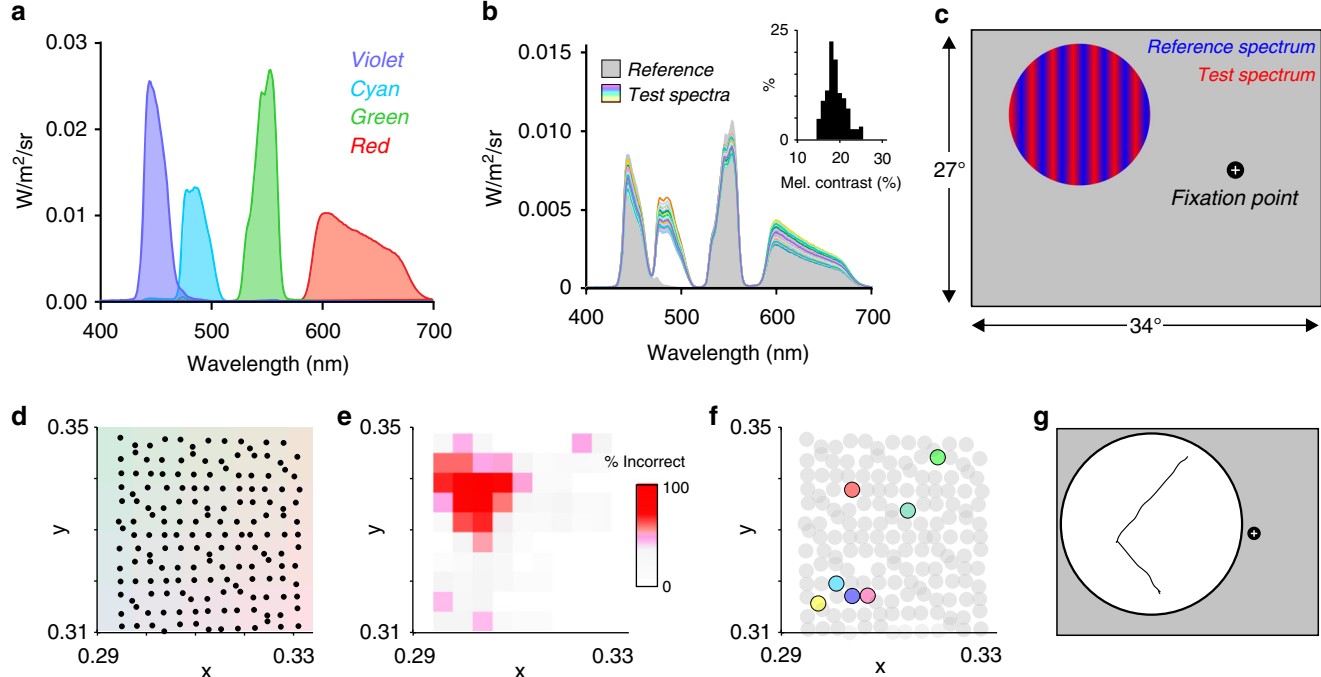

**Fig. 1** Psychophysical paradigm and online-tuning procedure used to identify metameric stimuli. **a** Spectral power distributions of the four primaries (violet, cyan, green and red). **b** Spectral power distributions of reference (grey) and test stimuli (coloured lines, lower panel) used to identify a metameric stimulus for each participant. Inset histogram plots the range of measured melanopsin contrasts between all reference and test spectra. **c** Diagram of stimulus presentation. Stimuli were presented on a projection screen occupying 27 × 34° of visual space. All participants were instructed to fixate on a cross with their left eye; all stimuli were presented within a circle (15° diameter), with centre located 20° from fixation point (6.5° above horizon; 19° in temporal direction). Sinusoidal gratings formed by mixing reference and test spectra in different ratios were presented in one of four orientations. **d** Predicted location of test stimulus spectra in *xy* colour space. Reference spectrum was held constant at $x = 0.31$ $y = 0.33$ (central point). **e** Heat map showing the proportion of incorrectly identified orientations for gratings formed by test and reference spectra as a function of the predicted *xy* coordinate of the test stimulus ($x = 0.31$ $y = 0.33$) for a representative participant. Note that incorrectly identified gratings cluster for test spectra in a particular region of colour space. **f** *xy* colour coordinate of the test stimulus (coloured circles) that was judged to be metameric to the reference stimulus (central point) for each participant. **g** Appearance of Purkinje tree sketched by one participant, viewing 3.2 cpd gratings with elevated penumbral L cone contrast, which were displayed over an enlarged visual space

We therefore applied the validated metamers to generate sinusoidal gratings over a broad range of spatial frequencies (0.2–12.8 cpd) and presented them to subjects in the four alternative forced choice orientation detection task. The subjective experience of the participants was that gratings at lower, but not higher, spatial frequencies could indeed be perceived, albeit as patterns that were 'hard-to-discern'. This was reflected in the objective grating orientation test, with subjects performing significantly above chance for spatial frequencies ≤0.8 cpd (Fig. 2a; one-sample *t*-test comparison with 25%, $p < 0.01$), but making occasional errors at all frequencies. For comparison, we repeated this test with gratings at moderate (11%) and low (2%) greyscale Michelson contrast or formed by selective application of contrast to each of the cone classes independently (Fig. 2a, b; Supplementary fig. 2). Importantly, all such cone-visible stimuli were readily detected at spatial frequencies above the threshold for metameric gratings (Fig. 2a; Supplementary fig. 2), indicating that the preference for very coarse gratings was a unique property of melanopsin-directed patterns.

We were able to reduce the melanopic contrast of the gratings by producing spectra intermediate between the two metamers (Fig. 2c). We applied this manipulation to gratings at spatial frequencies that were detected at our highest contrast (0.2–0.8 cpd) to describe contrast response functions across this range. As expected, there was a positive correlation between melanopsin contrast and the likelihood of correctly identifying grating orientations (Fig. 2d). The lowest detectible contrast was similar

across spatial frequencies (Michelson contrast of 16, 14 and 16% for 0.2, 0.4 and 0.8 cpd; one-sample *t*-test comparisons with 0.25, $p \leq 0.05$), indicating that sensitivity was fairly constant across this range of lower spatial frequencies and that the threshold for melanopsin-based discrimination lies around 15%.

Metamer pairs were designed to present contrast for melanopsin, but also retained a smaller contrast for rods (estimated 8% Michelson contrast). To test whether this rod contrast was the origin of grating detection we reapplied the stimuli at lower intensities. Increases in light intensity have opposite effects on the activity of rods vs melanopsin, with the former being most active under dim and the latter under bright light levels. Subjects were dark adapted for ≥20 min (consistent with ISCEV standards for measurement of dark-adapted responses[32]), and metameric patterns presented at 4000× lower intensity to place them well within the operating range of rods (0.054 cd m$^{-2}$). Subject performance at higher intensities was not reproduced at this dimmer background, with correct answers failing to reach statistical significance at any of the spatial frequencies tested (Fig. 3a). Interleaved greyscale gratings (11% Michelson contrast) at the dimmer background were correctly identified across moderate spatial frequencies (Fig. 3a). To further describe the intensity range over which the metameric patterns were detectable, we continued to describe the detection rate of greyscale and melanopic gratings at 0.4 cpd also at three intermediate intensities. Performance improved as a function of light intensity, but fell above chance only for the highest radiance

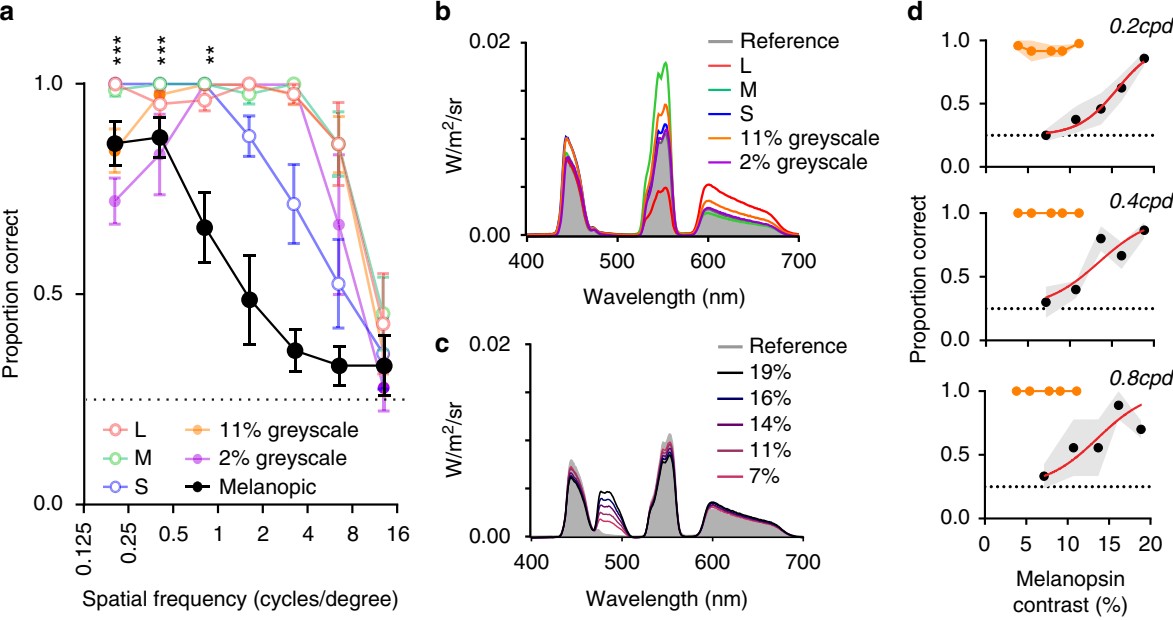

**Fig. 2** Spatial frequency tuning of melanopsin form vision. **a** Proportion of correctly identified stimuli at spatial frequencies between 0.2 and 12.8 cycles per degree (cpd), individually calibrated 'melanopic' stimuli (black) and stimuli designed to target cone photoreceptors (11% L, M or S, 11% greyscale, 2% greyscale, represented by red, green, blue, orange and purple points, respectively). Repeated measures one-way ANOVAs showed significant effect of spatial frequency for all sets of stimuli ($p < 0.001$). The proportion of correct responses at each spatial frequency was compared with chance (0.25) using a one-sample $t$-test. For melanopic stimuli: **$p < 0.01$; ***$p < 0.001$. Data show mean ± SEM; $n = 7$. Where no error bars are plotted, all participants were correct in 100% of trials. **b** Spectral power distributions of stimuli targeting cone photoreceptors. Grey shaded spectrum: background spectrum (common to all spectral pairs); coloured lines show the other element of a stimulus pair used to form gratings at 11% L, M or S Michelson contrast, 11% greyscale, 2% greyscale (red, green, blue, orange and purple points, respectively). **c** Spectral power distributions of stimuli targeting melanopsin at a range of contrasts (stimuli as calibrated for one participant). Grey shaded spectrum: background spectrum (common to all spectral pairs); coloured lines shows spectra used to present 7–19% melanopsin contrast. **d** Contrast response functions for melanopic (black) and greyscale stimuli (orange) for 0.2, 0.4 and 0.8 cpd stimuli. Grey shading shows SEM. Sigmoidal dose–response curves were fitted to data (red lines)

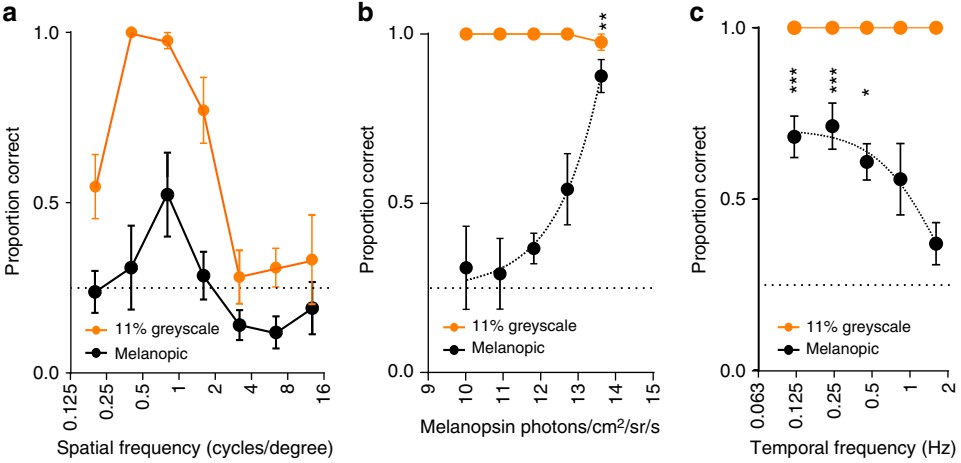

**Fig. 3** Defining the sensitivity limits of melanopsin form vision. **a** Proportion of correctly identified stimuli at spatial frequencies from 0.2 to 12.8 cpd using individually calibrated melanopic (black) and 11% greyscale stimuli (orange) with stimuli attenuated by ~4000 (mean luminance 0.054 cd/m²; black edged circles). Repeated measures one-way ANOVAs showed significant effect of spatial frequency for greyscale ($p < 0.001$) but not melanopic stimuli ($p = 0.09$). For melanopic stimuli, no responses at ND4 were significantly different from 0.25 (one-sample $t$-test). Data show mean ± SEM; $n = 7$. **b** Irradiance response function for melanopic (black) and 11% greyscale stimuli (orange) for 0.4 cpd stimuli. A repeated measures one-way ANOVA showed significant effect of intensity for melanopic stimuli ($p < 0.001$). Responses were compared using a one-sample $t$-test with a mean of 0.25 (chance point. **$p < 0.01$). Data show mean ± SEM; $n = 7$. Where no error bars are plotted, all participants were correct in 100% of trials. **c** Proportion of correctly identified stimuli presented at temporal frequencies of 0.125–2.5 Hz (spatial frequency of 0.4 cpd) using individually calibrated 'melanopic' stimuli (black) and 11% greyscale stimuli. A repeated measures one-way ANOVA showed significant effect of temporal frequency for melanopic stimuli ($p < 0.05$). Responses were compared using a one-sample $t$-test with a mean of 0.25 (chance point); for melanopic stimuli: *$p < 0.05$; ***$p < 0.001$. Data show mean ± SEM; $n = 5$. Where no error bars are plotted, all participants were correct in 100% of trials

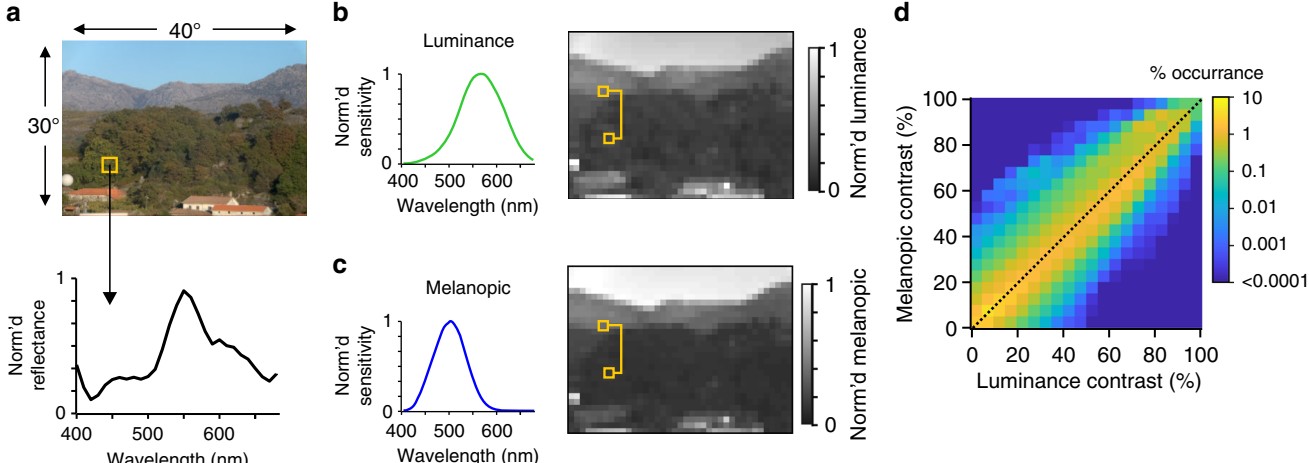

**Fig. 4** Divergence in melanopsin vs. luminance contrast in natural scenes. **a** Upper panel shows representative RGB rendering of a hyperspectral scene[43]. Images were assumed to occupy 30° × 40° visual space. Lower panel shows spectral power distribution for a representative 'pixel' from this scene (occupying a 1.5° square of visual space to approximate spatial resolution of melanopic vision). **b** Depiction of image in (**a**) spatially down sampled to produce 'pixels' at 1.5° square and with greyscale level used to depict luminance (scale bar to right, 0 and 1 = lowest and highest luminance pixel) calculated by applying the luminance sensitivity function[42] (shown to left) to spectral power distribution for each pixel. Yellow boxes indicate a representative pixel pair across which luminance contrast can be calculated. **c** As for (**b**), but depicting melanopic radiance calculated using melanopic sensitivity function (to left). **d** 2D histogram showing distribution of luminance and melanopic radiance contrast for all pixel pairs (as a % of total pixel pairs per image; scale to right) from 13 hyperspectral images. Dotted line shows point of equal luminance and melanopic contrast

stimuli (Fig. 3b; 214 cd m$^{-2}$). The relationship between light intensity and detectability revealed by these experiments is inconsistent with a rod origin for detection of metameric stimuli. As rods, cones and melanopsin are thought to be the only photoreceptors capable of providing visual information, the most parsimonious explanation for these data is that discrimination of these metameric gratings relies upon melanopsin. They further describe an intensity threshold in the region of 10$^{13}$–10$^{14}$ melanopsin photons cm$^{-2}$ sr$^{-1}$ s$^{-1}$ for this aspect of vision (at least for the moderate contrast stimuli employed here).

**Low temporal frequency bias of melanopsin vision.** Melanopsin-evoked responses are typically characterised by low temporal frequency bias. We used the validated metameric stimuli to establish whether this was true also for spatial discrimination. We presented gratings at a fixed size (0.4 cpd), inverting sinusoidally at frequencies ranging from 0.12 to 1.6 Hz (limitations of the projector system meant we were unable to test higher temporal frequencies). Thanks to the sinusoidal nature of the temporal modulation, these gratings should become intermittently apparent (at phases in the sinusoid in which melanopsin spatial contrast is above threshold) when presented at rates of change that melanopsin can track. Conversely, time averaging at higher frequencies should render the gratings invisible. We found that, indeed, subjects performed better than chance at reporting the orientation of these gratings only across lower temporal frequencies <0.8 Hz (Fig. 3c). This low frequency bias is consistent with the characteristics of melanopsin photoreception in other species[24] and with melanopsin contributions to regulating pupil size in humans[25].

**Melanopsin augments the appearance of greyscale patterns.** The metameric stimuli represent a powerful analytical tool to study 'melanopic' vision in isolation. Such, patterns visible only to melanopsin will be rarely, if ever, encountered in everyday life. Nevertheless, situations of divergent melanopsin and cone contrast are encountered. Thus, an analysis of a panel of hyperspectral images revealed that spatial contrast for melanopsin often

differs from luminance contrast (Fig. 4a–d; and other measures of light intensity (e.g. L + M + S excitation or total radiance, supplementary fig. 3)).

We next therefore adapted the metameric approach to address the question of whether melanopsin influences the appearance of patterns that are also visible to other photoreceptors. We considered first the special case in which coarse patterns may be at the limit of detectability for cones. The spatial frequency response curve constructed for greyscale stimuli (Fig. 2a) reveals that subjects made occasional errors when reporting the orientation of gratings at the lower two frequencies, even at contrasts that were readily detectable at higher frequencies. We asked whether melanopsin enhances detectability for such low spatial frequency patterns by using the metameric stimuli to add 20% melanopic contrast to the 2% greyscale gratings at 0.4 cpd. We found that the inclusion of melanopic contrast significantly enhanced detection over the simple greyscale stimulus, to the extent that subjects were now able to record grating orientation without error (Fig. 5a). Augmenting melanopic radiance did not improve detection of higher frequency grating (6.4 cpd) confirming that this effect was selective for lower spatial frequencies (two-way ANOVA main effect of spatial frequency ($p < 0.01$) and frequency × melanopsin contrast interaction ($p < 0.05$); post hoc Bonferroni multiple comparisons test between 2 and 2% + 20% melanopic, $p < 0.05$ at 0.4 cpd; $p = 0.71$ at 6.4 cpd). These data indicate that melanopsin can improve detectability of coarse patterns.

In order to determine the potential for melanopsin to impact pattern appearance over a wider array of conditions, we next generated a range of 0.4 cpd greyscale gratings varying in Michelson contrast from 6 to 33% between brightest and dimmest pixel, and a parallel set of stimuli in which inclusion of the validated metamers enhanced melanopsin contrast by 17–20% (Fig. 4d). Gratings were clearly apparent in each member of the resultant stimulus pairs and, importantly, differences between cone and melanopic contrast in the augmented melanopsin contrast stimuli were within the range encountered in hyperspectral images of natural scenes (9.1% of pixel pairs in our hyperspectral image panel for which luminance contrast is

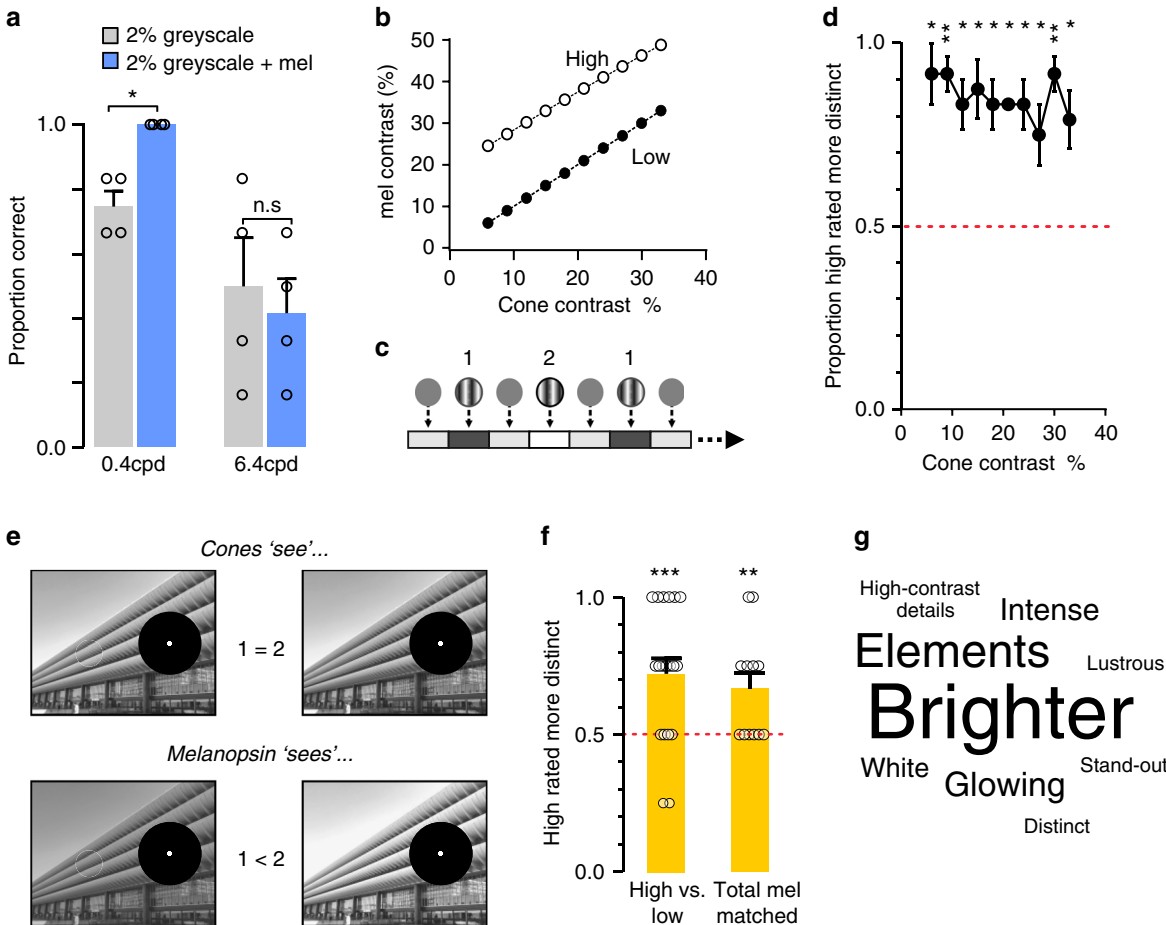

**Fig. 5** Melanopsin contrast can be used to detect and distinguish spatial patterns. **a** Proportion of correctly detected 0.4 and 6.4 cpd gratings, for 2% greyscale (grey bars) and 2% greyscale +20% melanopic (blue bars) spectra. Data were compared with two-way ANOVA, finding a significant effect of spatial frequency ($p < 0.01$) and the interaction between spatial frequency and spectrum ($p < 0.05$). A post hoc Bonferroni multiple comparisons test compared spectral condition at the two frequencies (*$p < 0.05$). Data show mean ± SEM; $n = 4$. **b** Stimulus contrast for L, M and S cones ($x$ axis) and melanopsin ($y$ axis), for high and low-melanopic stimuli (white and black, respectively). **c** Two-alternative forced choice stimulus presentation protocol. In all cases, two stimuli were presented in series until a participant responded. In each case, stimuli presented the same cone contrast, but differed in their melanopic contrast. **d** Proportion of times participants selected the high-melanopic contrast stimulus as being more 'distinct'. Responses were compared using a one-sample $t$-test with a mean of 0.5 (chance point; *$p < 0.05$; **$p < 0.01$). Data show mean ± SEM; $n = 4$. **e** Cartoon depicting stimulus presentation of greyscale images. Images were presented across the projection screen with 10° surrounding fixation point obscured. Greyscale images were presented using metameric low and high-melanopic settings (versions 1 and 2 in figure, respectively). On a second trial, images were also matched in mean melanopic radiance (but varied in the range of melanopic radiances presented across the image (i.e. high vs. low-melanopic spatial contrast)). **f** Bar graph depicting proportion of times participants ($n = 4$) selected the high-melanopic image as being more 'distinct' for low- and high-melanopic images. Participants showed significant preference for high-melanopic images, even when images were matched in their mean melanopic radiance ('Total mel matched'). Responses were compared using a one-sample $t$-test with a mean of 0.5 (chance point; **$p < 0.01$; ***$p < 0.001$). Data show mean ± SEM; circles show mean rating for individual images. **g** Word cloud generated using four participant's free-form descriptions of high-melanopic images compared with low-melanopic image. Word size relates to number of times participants used this descriptor to describe high-melanopic images

6–33% differ by >20% in melanopic vs. luminance contrast). Subjects were presented with these stimuli in pairs matched for greyscale (and therefore cone) contrast and asked to identify the pattern that looked 'more distinct' (paradigm shown in Fig. 5c). In this task, participants preferably selected the pattern with an elevated melanopsin contrast across the range of cone contrasts tested (Fig. 5d; one-sample $t$-test comparison with 50%, $p \leq 0.05$). This preference indicates that melanopsin alters the appearance of coarse patterns even when accompanied by quite large contrasts visible to cones.

**Enhancing melanopsin contrast alters image appearance.** The experiments with gratings demonstrate that melanopsin can contribute to the appearance of low frequency patterns across a

range of moderate contrasts. Such patterns are commonly encountered in everyday life and these findings therefore imply that melanopsin can influence the appearance of typical scenes. We finally tested this hypothesis by using our calibrated metameric pairs to vary the melanopsin contrast of greyscale representations of 19 everyday scenes.

Members of each pair were then presented sequentially in a randomised order (image presented for 2 s, interleaved with 1 s mid-intensity 'grey' background). To avoid confounds with foveal vision, subjects fixated on a cross and a 10° region surrounding this fixation point was covered by a grey mask (Fig. 5e). Images were presented in series, and participants were asked to select in which of the two versions of the image 'patterns were more distinct'. Participants showed a preference for selecting high-melanopic images, consistent with the hypothesis that

augmenting melanopsin contrast had a reproducible impact on image appearance (Fig. 5f; one-sample $t$-test comparison with 50%, $p$ 1< 0.001).

In addition to varying in melanopsin spatial contrast, these first set of low vs high-melanopic images had modest differences also in overall melanopic radiance (mean ± SD Michelson contrast in overall melanopic radiance = 11.5 ± 1.7%). To preclude the possibility that subjects were actually responding to those differences in global melanopic radiance (rather than differences in the spatial pattern of melanopic intensity) we produced a second set of metameric image pairs differing in melanopic spatial contrast but not in mean melanopic radiance ($n$ = 12; see the Methods section). We again asked participants to select in which version of these 'total melanopic matched' images 'patterns were more distinct', and found that participants retained their preference for images with high-melanopic contrast (Fig. 5f; one sample $t$-test comparison with 50%; $p$ < 0.01).

To better describe the nature of this impact, we selected those images in which 100% of participants selected the high-melanopic image, and asked viewers to provide free-form descriptions of the differences between the two versions. Viewers consistently described the high-melanopic image (and/or elements within it) as appearing brighter, more 'stand-out', glowing and higher-contrast. A word cloud summarises the descriptors that participants used to describe how the higher-melanopic images appeared (Fig. 5g).

## Discussion

We present a method for selectively enhancing spatial contrast for melanopsin and apply it to explore the contribution of melanopsin to pattern vision in healthy human subjects. We find that 'melanopic' patterns (with spatial contrast for melanopsin but not cones) are visible at low spatial and spatiotemporal frequencies and at higher background light intensities. The contrast sensitivity of this 'melanopic' vision is low, compared with rods and cones, but well within the range typically encountered in everyday life. Selectively enhancing melanopsin contrast alters the appearance of greyscale gratings and a selection of everyday images, making patterns appear more distinct and image elements 'brighter'. These data are consistent with the hypothesis that melanopsin can be used to detect and distinguish spatial patterns and that this inner-retinal photoreceptor makes a direct contribution to form vision.

Receptor silent substitution and metamerism are well-established principles of colour science and grounded in fundamental photoreceptor properties (that each receptor has the same unitary response to light of different wavelength, but is differentially sensitive across the spectrum[33]). We use these methods here to generate pairs of spectrally distinct lights that differ in effective intensity for melanopsin but are matched for each of the L, M and S cones (metamers). The resultant ability to modulate melanopsin activity selectively represents the only realistic opportunity to study melanopsin's contribution to vision in healthy humans. However, its utility is critically dependent upon patterns generated with the metamers indeed being visible only to melanopsin and not cones (or rods). Several aspects of our data provide confidence that this is indeed the case in our study. Addressing first the possibility of inadvertent cone contrast: we minimised the possibility of this in our experimental design by using an online-tuning step to identify metameric pairs independently for each subject that were invisible when presented in a pattern to which cones are very sensitive (3.2 cpd). Metamerism can fall down in the shadow of blood vessels[30], and we further designed our stimuli to have little contrast for such penumbral cones. Accordingly, our subjects did not detect metameric

patterns at high spatial frequency (as expected for penumbral cone contrast; ref. [30]) nor report seeing a Purkinje tree of retinal vasculature. The range of spatial and spatiotemporal frequencies over which the metameric stimuli were detected is also inconsistent with the hypothesis of residual cone contrast. Metameric stimuli were invisible across a range of frequencies at which cone vision is most sensitive, and detectable only at low frequencies.

Although our metameric stimuli were designed to minimise rod contrast, we were unable to completely eliminate it without also substantially reducing melanopsin contrast (as these receptors have fairly similar spectral sensitivities). Several aspects of our data, however, provide confidence that detection of metameric patterns relies upon melanopsin rather than rods. Firstly, metameric patterns were invisible at a low light level, at which rod vision should be active, and only became detectable at the brightest background tested. While, rods can be active at high light levels[34], we are unaware of a precedent for rod-based vision becoming more influential across such a mesopic to low photopic transition. Secondly, metameric patterns were not detectable at spatial and spatiotemporal frequencies that rods are very sensitive to (e.g. at spatial frequencies >1 cpd and spatiotemporal frequencies >1 Hz, which are readily tracked in scotopic conditions and achromats[35,36]).

The sensory properties of melanopsin vision revealed by the metameric patterns are consistent with descriptions of melanopsin activity in other contexts. The threshold temporal frequency for metameric pattern detection (<1 Hz) is similar to that reported for electrophysiological responses in the mouse dLGN[24] and for melanopsin-directed pupil responses in humans[25,28]. Similarly, the threshold light intensity for detecting melanopsin responses is consistent with previous in vivo and in vitro measures[2,3,10]. To our knowledge, the only other explorations of spatial resolution for melanopsin photoreception have been undertaken in mice[24], and differences in eye size preclude simple comparisons across species. Nonetheless, in those cases as well as this study, the threshold spatial resolution is around that expected based upon mRGC anatomy. The contrast sensitivity of melanopsin vision (threshold around 14% Michelson contrast) is modest compared with that of rods and cones, but is consistent with previous reports based upon pupil responses[14]. It is higher than that determined for electrophysiological responses in the mouse dLGN[13,24], but it is feasible that limits of detection are superior for psychophysical compared with electrophysiological readouts.

The characteristics of melanopic vision (low spatiotemporal resolution and contrast sensitivity) preclude it from contributing to high spatial acuity vision. Nevertheless, the types of patterns to which melanopsin would be sensitive are common in everyday scenes. This raises the question of how melanopsin vision may contribute to image appearance. We find that subjects are able to distinguish greyscale patterns with augmented melanopsin contrast from those without. Consistent with previous descriptions of the experience of melanopsin-directed full-field and featureless stimuli, subjects record stimuli with augmented melanopsin contrast as differing in percepts related to 'brightness'. Thus, subjects reliably reported greyscale patterns (gratings and images) with enhanced melanopsin contrast as 'more distinct'. Moreover, when asked to describe the difference between regular and melanopsin-augmented greyscale images, subjects used adjectives such as 'brighter', 'glowing' and more 'stand out'. In at least some ways then, the experience of additional melanopsin contrast is similar to that of enhanced luminance contrast. Previous studies based upon assessments of full-field stimuli or ambient lights have suggested that perceived 'brightness' is produced by a sum of luminance and melanopsin signals[22,37]. However, these have not gone so far as showing that there is no perceptual consequence

for substituting luminance with melanopic radiance, and the experience of temporal modulations visible only to melanopsin has been described as quite peculiar[14]. We find that at the limits of detection, including melanopsin contrast can compensate for low energy contrast (Fig. 5) but as the inclusion of chromatic contrast can have a similar effect, this is not good evidence of equivalence. Moreover, although subjects in this study were able to perform substantially better than chance at reporting the orientation of metameric gratings at low spatial frequencies, their subjective experience was that these did not look equivalent to low contrast (hard to discern) greyscale stimuli. While the latter gave a clear percept of oriented bars (when detectable), in many cases metameric gratings rather gave a more diffuse impression of a pattern, without individual elements being clearly visible per se. Thus, an important direction for future work will be to determine whether melanopsin is interchangeable with luminance contrast at the perceptual level, or supports a distinct percept related to 'brightness'. Similarly, it will be important to ascertain how altering melanopsin contrast impacts the appearance of colour patterns and the detectability of features other than grating orientation, such as object size and shape.

## Methods

**Participants and eligibility criteria.** Seven observers (four males, three females) were recruited from the wider University of Manchester community. All subjects had 6/5 or better visual acuity and normal red-green colour vision as assessed by the Ishihara test[38]. Two participants were authors of this study (A.A. and R.J.L.), they did not contribute data to the free-form descriptions of image appearance presented in Fig. 5.

**Visual stimuli.** Stimuli were presented using a projection system similar to that first described in ref. [27]. Briefly, we modified the output of two projectors by inserting interference filters into the light path of the blue and green channels of each. Thus, in a first projector, the blue primary was modified using a 470 nm cut-off yellow longpass filter (PIXELTEQ, Largo, FL; part # LP470-r40 × 25 × 1), making a 'cyan' primary. In a second projector, the blue primary was modified using a 463–571 nm magenta notch filter (MidOpt, Palatine, IL; part #102340892), making a 'violet'; and the green primary was modified using a 550 nm bandpass filter (PIXELTEQ, Largo, FL part # Bi550-r40 × 25 × 1), producing a narrower green primary. By superimposing the output of these two projectors (including the un-filtered red channel from projector 2), we achieved a four primary display. Projectors were controlled independently with a PC running Processing v.3.3.1 (Processing Foundation).

**Photoreceptor specific modulations.** A spectroradiometer (SpectroCAL MKII; Cambridge Research Systems, UK) was used to calibrate the output of the projector channels individually on an 8 bit scale. Following this, a range of stimuli were generated to modulate the activity of targeted photoreceptors. LMS cone spectral sensitivities were based upon CIE 2006 10° photoreceptor sensitivities[29]. The spectral sensitivities of melanopsin and rod opsin were approximated using visual pigment templates with λ = 480 and λ = 498, respectively[39], corrected for pre-receptoral filtering as for LMS. Penumbral cone contrasts were also calculated, using the spectral absorption profile of haemoglobin[30].

All photoreceptor-targeting stimuli were generated relative to a 'white' background stimulus (CIE 1931 xy chromaticity: (0.31, 0.33)) with a luminance of 214.4 cd m$^{-2}$. This had L, M, S, Melanopic and rod log effective photons cm$^{-2}$ s$^{-1}$ sr$^{-1}$ of 14.0; 13.9; 13.6; 13.7 and 13.6, respectively. In some instances, neutral density filters were placed in front of both projectors to provide a spectrally neutral reduction in intensity in 10$^{0.9}$ increments. The spectra of all stimuli used were all measured with a spectroradiometer, and relevant contrasts are described below where relevant.

**Procedure overview.** In all stimulus protocols, stimuli were presented in a fully darkened room. Participants were seated 148 cm from a front projection screen, onto which the superimposed output of the projectors was displayed over an area of 109 × 158 cm. Participants were positioned such that the projected image covered a total visual angle of 34 × 27 degrees. See supplementary fig. 1 for geometry. Participants were asked to fixate on a cross with their left eye (right eye masked). Following this, a circular stimulus (15° diameter) was presented at a fixed retinal eccentricity (centre: 19° temporal, 6.5° superior). The location of the stimulus avoided overlap with the blind-spot. For sections of the study, we monitored the accuracy of participant's fixation. Participants wore an eye tracking headset (Pupil Labs GmbH, Berlin, Germany), monitoring gaze direction of the left eye (right eye-covered during experiment) and a forward facing camera to cross reference eye

gaze with a participant's view. Using software provided by the manufacturer, fixation was validated during the online-tuning protocol. Using this general set up, several protocols were undertaken which are described in detail below.

**Online-tuning protocol.** To identify a spectral pair which were functionally metameric for each participant, a range of 170 spectra were presented as a spatial sinusoidal grating at a fixed spatial frequency (see Fig. 1). Stimuli were presented in one of four orientations, and participants were then asked to report which of four orientations gratings were presented within the circular stimulus (4-alternative forced choice paradigm; 25% false-positive rate). Each stimulus was displayed until a participant reported the orientation. Following this, the stimulus disappeared (replaced with homogenous background) for a period of 2 s, following which the next stimulus was presented. Grating orientation was fully randomised. All spectral pairs were presented twice in a pseudorandom order, with the goal of identifying stimuli that the participant was unable to discriminate the orientation of gratings. Stimuli that were incorrectly identified at least once were presented a third time. This allowed us to identify (at least one) spectral pair that was incorrectly detected 3/3 times, for each participant. For all our participants, this process was successful in identifying a 'functionally metameric' stimulus.

**Spatial frequency testing protocol.** Stimuli were presented at one of six spatial frequencies, and were displayed until a participant reported their answer, with an inter-stimulus interval of 2 s. Stimuli were displayed in a pseudorandom order at a range of contrasts, using stimuli targeting melanopsin (using user-specific calibration setting), an equal-energy change in spectrum ('greyscale') at one of two contrasts (11 and 2% Michelson contrast for all photoreceptors) and targeting L, M and S cones (11% Michelson contrast for selected photoreceptor). Note that LMS targeting stimuli are based on the CIE 10° cone fundamentals based on the standard observer (not individually calibrated). Each stimulus was presented six times, and the % correct responses calculated. The % of correct responses for each contrast was then compared with a level of chance (25%) using a one-sample $t$-test.

**Contrast sensitivity protocol.** To reduce the contrast of melanopsin-targeting and greyscale stimuli, we generated spectra that were intermediate between the background and photoreceptor-targeting stimuli. These stimuli of varying contrasts were displayed in a pseudorandom order, using stimuli targeting melanopsin (using user-specific calibration setting) or an equal-energy change in spectrum, 'greyscale' (targeting all photoreceptors). As previously, stimuli were displayed until a participant reported their answer, with an inter-stimulus interval of 2 s. Spatial frequencies from 0.2–0.8 cpd were tested. Each stimulus was presented six times, and the % correct responses calculated. The % of correct responses for each contrast was then compared with a level of chance (25%) using a one-sample $t$-test. The threshold for contrast detection was plotted as a contrast sensitivity function, taking the reciprocal of contrast threshold and plotting as a function of spatial frequency.

**Temporal frequency protocol.** Temporally modulated sign inverting gratings were presented at frequencies from 0.125 to 1.6 Hz. A background disc (intermediate intensity spectrum) was first presented for 2 s. Participants were then cued with a presentation of a white circle in their fixation point, following which one cycle of an inverting sinusoidal grating was presented within the stimulus disc. Following this, the intermediate intensity background disc was presented until a participant reported their answer. As previously, participants were asked to report the orientation of the inverting grating. Stimuli were displayed in a pseudorandom order, using stimuli targeting melanopsin (using user-specific calibration setting) or greyscale stimulus (11% Michelson contrast for all photoreceptors). A fixed spatial frequency of 0.4 cpd was used. Each stimulus was presented six times, and the % correct responses calculated. The % of correct responses for each contrast was then compared with a level of chance (25%) using a one-sample $t$-test. Note that the potential sensitivity to lapses in attention of a task involving detection of intermittent stimuli, especially at low temporal frequencies, preclude further conclusions regarding the optimum integration time for melanopsin vision from these data.

**Two-alternative forced choice discrimination protocol.** Pairs of gratings were generated in which the higher intensity component of the grating was either the background spectrum or an individual's calibrated metameric spectrum; and the lower intensity component of the grating was the same within each pair. The lower intensity component was adjusted to present a range of cone contrasts. This approach thus provided two versions of a range of cone stimuli; one in which the melanopsin contrast was equal to the cone contrast, and a second in which the melanopsin contrast was enhanced. Each stimulus was presented for 1 s, separated by a 1 s inter-stimulus interval (see Fig. 5), for three repeats. The order of presentation was randomised. Gratings were presented in the same orientation for each pair of gratings, but varied randomly one from stimulus pair to the next. Participants were then asked to decide in which of the two stimuli the pattern was more distinct.

Two versions of pairs of greyscale images (selected from a range of everyday scenes) differing in melanopic contrast were produced. In the first, the brightest point in each version of an image was the set to be one of the calibrated metameric stimuli (melanopsin bright or dim) for a participant. Spectrally neutral reductions

in intensity were then used to generate greyscale values for all other pixels in the image. Using these values, we were able to generate 19 metameric greyscale images, differing in melanopic spatial contrast, but also in overall melanopic radiance. In a second version, we matched image pairs for mean melanopic radiance, by first defining the median intensity pixel within each image. For the high-melanopic contrast images, we rendered all greyscale values below this point in the low-melanopic metamer, and those above it with the high-melanopic metamer. For the low-melanopic contrast images, we did the reverse. This approach allowed us to generate 12 images with varying melanopic contrast, but matched for mean melanopic radiance.

Each image was presented for 2 s, separated by a 1 s inter-stimulus interval, for as many repeats as the participant wished to view. Images were displayed across the entire projection area. To avoid confounds with foveal vision, subjects fixated on a cross and a 10° region surrounding this fixation point was covered by a grey mask. The order in which the two versions of any image were presented was randomised. Participants were then asked to decide in which of the two images 'patterns were more distinct'. In a follow-up experiment, the same participants were asked to view those images ($n = 6$) in which the higher-melanopic image was rated as more distinct in 100% of trials. In this case, however, the high-melanopic image was always presented second in the sequence. Participants were then simply asked to describe how the second image appeared compared with the first. A word cloud was then generated by scaling the font of each word according to the number of times a descriptor was used.

**Hyperspectral image analyses.** Hyperspectral images[40,41] were used to estimate how melanopsin covaries with luminance and other measures of photoreceptor activity when viewing natural scenes. To generate this information, hyperspectral images were assumed to locate an area of 30° × 40°, and then spatially filtered for an array of hypothetical non-overlapping mRGCs with receptive fields 1.5° diameter. Spectral reflectance data from each location was then transformed into reflected radiance data relative to luminance[42], or L, M, S and melanopsin photoreceptors using nomograms as described above for spectral data[39]. The spatial Michelson contrast between all pairs of receptive fields was then computed, to provide information about luminance, L, M, S and melanopsin contrast presented within each image. Those data were then summarised in two-dimensional histograms plotting relative contrasts for different photoreceptors, between all pixel pairs. Computations were performed in MATLAB (R2017a).

**Ethics.** Ethical approval was obtained from the University of Manchester Ethics commission (approval number #2017-2276-3181). Informed consent was obtained from all participants.

**Reporting summary.** Further information on research design is available in the Nature Research Reporting Summary linked to this article.

## Data availability
The data that support the findings of this study are available from the authors on reasonable request. The source data underlying Figs. 2, 3, 5a, d, and f are provided as a Source Data file.

## Code availability
Wherever possible, codes used to generate stimuli are available on request from the authors.

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

## Acknowledgements

The authors would like to thank Dr Philip Wright, Richard McDowell and Marina Gardasevic for their assistance with data collection, and thank Dr Sei-Ichi Tsujimura, Dr Manuel Spitschan and Dr Tom Woelders for comments on the manuscript. This work was supported by a Dean's Prize Fellowship funded by the School of Biological Sciences, University of Manchester, to A.E.A., a BBSRC project grant (BB/P009182/1 to R.J.L.), and Wellcome Trust Investigator Award to R.J.L.

## Author contributions

A.E.A. and R.J.L. designed the study, apparatus and experiments. A.E.A. performed experiments and analysed data. F.P.M. produced the multiprimary display device. A.E.A. and R.J.L. wrote the manuscript, the final version of which was which was read and approved by all authors

## Additional information

**Competing interests:** A.E.A. and R.J.L. are listed as inventors on patent PCT/GB2017/050338: 'Improvements in image formation'. F.P.M. declares no competing interests.

