## [Peer Review File · Nature Communications]

Reviewers' comments:

Reviewer #1 (Remarks to the Author):

Manuscript#: NCOMMS-18-38364

Corresponding Author: Annette E Allen

Title: Form vision from melanopsin in humans

This paper investigates the contribution of inner retinal photoreceptors (i.e. melanopsin containing retinal ganglion cells (mRGCs)) to form vision in human subjects. Traditionally, rod and cone photoreceptors in the outer retina have been viewed as the only neural populations involved in retinal image formation. However, the discovery of retinal ganglion cells which contain the photosensitive pigment melanopsin has forced a re-appraisal of this view. As yet a definitive characterisation of the exact nature of the contribution made by these inner retinal neurons to visual function – something this paper attempts to address.

In an initial set of experiments the authors employed a carefully designed and selected set of stimuli that stimulated either mRGCs or cone photoreceptor populations. Using these stimuli in psychophysical experiments they were able to characterise the spatial and temporal conditions under which the mRGCs contributed to their detection which was primarily apparent for patterns of low spatial and/or temporal frequency.

In a second series of experiments the authors go on to demonstrate that the addition of melanopsin contrast to a selection of real-world images altered their appearance to normal observers who rated them as more distinct/brighter than images that lacked this additional melanopsin contrast.

The findings are novel in that, as far as I am aware, this is the first time that the basic spatial and temporal characteristics of mRGCs contribution to form vision have formally been assessed in the human visual system. Moreover, this assessment has been undertaken with a great deal of care in terms of the stimulus design and experimental methodology. Silent substitution techniques have been employed to generate stimuli with melanopsin contrast that are 'invisible' to cones – but the authors are also aware of the drawbacks of silent sub stimulation (e.g. variations in observer cone spectral sensitivities, issues with penumbral cones, etc) and have taken appropriate steps to minimise contributions to detection from inadvertently stimulated cones (and rods). This tends to make the work in my view highly convincing.

There is perhaps one small aspect that I would like clarification on. The methods section (page 7, line 312) states that 7 subjects took part in the study and two of these were the authors (AA and RJJ). My reading of the data in fig 3 is that it is based in $n = 4$ subjects. I would just like re-assurance that this $n=4$ did not include the 2 authors - which would obviously raise issues of bias in the 'rating' experiments. Apart from this minor point of clarification I have no other issues with what I feel is a well-conceived, expertly executed and well-written study.

Declan McKeefry

Reviewer #2 (Remarks to the Author):

This manuscript describes a potential role for melanopsin in human spatial/form vision. The manuscript is well-written, the experimental design is clever and thorough, and the results are convincing. My primary concern is that the authors may be overstating the role of melanopsin in human form vision under non-laboratory conditions.

That is, the authors show convincingly that melanopsin activation alone is sufficient to mediate grating orientation judgments and that increasing melanopsin contrast can make low spatiotemporal frequency gratings more visible. However, as the authors point out, the natural environment does not contain stimuli that activate melanopsin in isolation. Likewise, it is unlikely that natural scenes would generate a melanopsin response that is 10x greater than the cone response. Consequently, melanopsin may be capable of supporting form vision in humans, but only under very unusual (lab-based) circumstances.

The experiments that employed natural images showed that subjects can differentiate between scenes that contain high versus low melanopsin contrast, but this is not evidence that melanopsin improves or mediates form vision. Rather, the results of these experiments suggest that melanopsin activation increases the sensation of brightness, which is likely to be of limited value under the high photopic conditions that best activate melanopsin. This is not a new finding, as the role of melanopsin in signaling overall luminance ("brightness") is known.

In sum, the most novel finding (melanopsin can mediate orientation discriminations) is likely relevant only under carefully designed lab settings, whereas the more impactful finding (melanopsin can alter brightness perception in natural images) is not particularly novel. In addition to concerns about overall impact, there were a few peculiarities in Fig. 2 that the authors may want to address:

- 1) Fig. 2e: Increasing the spatial frequency from 1.0 to ~1.6 cpd resulted in a sharp contrast sensitivity drop (easily visible to invisible). This is distinct from the typical CSF and does not seem physiologically plausible. Some comment would be helpful to interpret this result.
- 2) Fig. 2f: The authors state "Subjects were unable to reliably report the orientation of metameric gratings at any of the spatial frequencies tested..." However, subjects achieved 50% correct performance at 1 cpd. This seems to meet the definition of "reliable." Is the non-statically significant result due to low power (small N)?
- 3) Fig. 2h: For unlimited exposure duration, subjects achieved approximately 90% correct performance for the melanopic stimuli (Fig. 2a), whereas they only achieved approximately 70% correct performance for an 8 sec exposure duration (Fig. 2h). Note that the function in Fig. 2h asymptotes at 4 sec, which would not predict improvement for longer durations. This discrepancy would be apparent if the authors plotted the unlimited duration data on Fig. 2h (e.g. at minus infinity Hz). Do the authors propose that melanopsin has an integration duration of more than 8 sec? If so, it would be helpful to see measurements for longer exposure durations. For the unlimited duration condition, did the subjects typically view the stimulus for more than 8 sec before making a response?

Reviewer #1 (Remarks to the Author):

Manuscript#: NCOMMS-18-38364

Corresponding Author: Annette E Allen

Title: Form vision from melanopsin in humans

This paper investigates the contribution of inner retinal photoreceptors (i.e. melanopsin containing retinal ganglion cells (mRGCs)) to form vision in human subjects. Traditionally, rod and cone photoreceptors in the outer retina have been viewed as the only neural populations involved in retinal image formation. However, the discovery of retinal ganglion cells which contain the photosensitive pigment melanopsin has forced a re-appraisal of this view. As yet a definitive characterisation of the exact nature of the contribution made by these inner retinal neurons to visual function – something this paper attempts to address.

In an initial set of experiments the authors employed a carefully designed and selected set of stimuli that stimulated either mRGCs or cone photoreceptor populations. Using these stimuli in psychophysical experiments they were able to characterise the spatial and temporal conditions under which the mRGCs contributed to their detection which was primarily apparent for patterns of low spatial and/or temporal frequency.

In a second series of experiments the authors go on to demonstrate that the addition of melanopsin contrast to a selection of real-world images altered their appearance to normal observers who rated them as more distinct/brighter than images that lacked this additional melanopsin contrast.

The findings are novel in that, as far as I am aware, this is the first time that the basic spatial and temporal characteristics of mRGCs contribution to form vision have formally been assessed in the human visual system. Moreover, this assessment has been undertaken with a great deal of care in terms of the stimulus design and experimental methodology. Silent substitution techniques have been employed to generate stimuli with melanopsin contrast that are ‘invisible’ to cones – but the authors are also aware of the drawbacks of silent sub stimulation (e.g. variations in observer cone spectral sensitivities, issues with penumbral cones, etc) and have taken appropriate steps to minimise contributions to detection from inadvertently stimulated cones (and rods). This tends to make the work in my view highly convincing.

There is perhaps one small aspect that I would like clarification on. The methods section (page 7, line 312) states that 7 subjects took part in the study and two of these were the authors (AA and R.J.L). My reading of the data in fig 3 is that it is based in $n = 4$ subjects. I would just like re-assurance that this $n=4$ did not include the 2 authors - which would obviously raise issues of bias in the ‘rating’ experiments. Apart from this minor point of clarification I have no other issues with what I feel is a well-conceived, expertly executed and well-written study.

Declan McKeefry

We are grateful for this thoughtful appraisal of our manuscript. In response to the reviewer's specific question about the free-form descriptions of image appearance we are happy to confirm that none of the authors of the study contributed to those assessments. We have changed the methods section to make this clear: "Two participants were authors of this study (A.A. and R.J.L), they did not contribute data to the free-form descriptions of image appearance presented in figure 3."

Reviewer #2 (Remarks to the Author):

This manuscript describes a potential role for melanopsin in human spatial/form vision. The manuscript is well-written, the experimental design is clever and thorough, and the results are convincing. My primary concern is that the authors may be overstating the role of melanopsin in human form vision under non-laboratory conditions.

That is, the authors show convincingly that melanopsin activation alone is sufficient to mediate grating orientation judgments and that increasing melanopsin contrast can make low spatiotemporal frequency gratings more visible. However, as the authors point out, the natural environment does not contain stimuli that activate melanopsin in isolation. Likewise, it is unlikely that natural scenes would generate a melanopsin response that is 10x greater than the cone response. Consequently, melanopsin may be capable of supporting form vision in humans, but only under very unusual (lab-based) circumstances.

We agree that this is an important issue, and are grateful to this referee for identifying an area for improvement in our initial submission. As a point of clarification, we used 10x greater melanopsin contrast only when considering the special case of very low cone contrast gratings (results shown in Fig 3e). It was in order to extend our work to a wider range of more realistic stimuli that we continued to explore stimuli with higher cone contrast and a smaller difference between cone and melanopsin contrast (typically around 20%). The outcome of that work is shown in Fig 3h and we consider those data to represent better evidence for the utility of melanopsin under more commonly encountered circumstances. The differences in cone vs melanopsin contrast employed in the stimuli for Fig 3h are fairly commonly encountered in our panel of hyperspectral images. We apologise for omitting that critical information in our initial submission. In our resubmission, we have included a new panel in figure 3 (figure 3d) summarising the correlation between luminance contrast and melanopic contrast in the hyperspectral images (after an approximation of appropriate spatial filtering). We have also changed the text to make our motivation for undertaking that portion of the study clearer and to ensure that the link to our conclusions is more explicit, as follows: “The metameric stimuli represent a powerful analytical tool to study ‘melanopic’ vision in isolation. Such, patterns visible only to melanopsin will be rarely, if ever, encountered in everyday life. Nevertheless, situations of divergent melanopsin and cone contrast are encountered. Thus, an analysis of a panel of hyperspectral images revealed that spatial contrast for melanopsin often differs from luminance contrast (figure 3a-d; and other measures of light intensity (e.g. L+M+S excitation or total radiance, supplementary figure 2)).”

We have also altered the subsequent description of the grating stimuli in the results section to highlight their relationship to naturally occurring contrasts.

The experiments that employed natural images showed that subjects can differentiate between scenes that contain high versus low melanopsin contrast, but this is not evidence that melanopsin improves or mediates form vision. Rather, the results of these experiments suggest that melanopsin activation increases the sensation of brightness, which is likely to be of limited value under the high photopic conditions that best activate melanopsin. This is not a new finding, as the role of melanopsin in signaling overall luminance (“brightness”) is known.

We agree with this reviewer that, based upon the literature it is expected that melanopsin may contribute to an overall sensation of scene brightness (although note that this has not been previously demonstrated for patterned images). According to those data one would expect changes in the overall melanopic content of an image to be apparent, but there is no published indication that changes in the spatial distribution of melanopic irradiance could be perceived (other than our own electrophysiological study in mice - see introduction). We appreciate that our previously presented data did not make this distinction sufficiently clear. To address this, we have re-generated the natural images in which we varied melanopic brightness within the scene, with the critical difference that now the mean melanopic brightness across the image is matched across each image pair - only the spatial distribution of melanopic radiance within the image differs. Thus 'overall luminance ("brightness")' is invariant for each image pair. They differ only in spatial contrast for melanopsin. We believe that this is a good way of separating an overall percept of 'brightness' (as might occur when the overall melanopic radiance of each image differed) vs. patterns in radiance. We find that indeed image appearance is altered by this selective manipulation of melanopsin spatial contrast. These data are now presented in figure 3j and within associated text.

In sum, the most novel finding (melanopsin can mediate orientation discriminations) is likely relevant only under carefully designed lab settings, whereas the more impactful finding (melanopsin can alter brightness perception in natural images) is not particularly novel.

We hope that the responses above address this concern. In addition, we have changed the final sentence of the abstract to introduce appropriate caution in our conclusion. This sentence now reads: 'These data identify melanopsin as a new potential origin for aspects of spatial vision in humans.'

In addition to concerns about overall impact, there were a few peculiarities in Fig. 2 that the authors may want to address:

1) Fig. 2e: Increasing the spatial frequency from 1.0 to ~1.6 cpd resulted in a sharp contrast sensitivity drop (easily visible to invisible). This is distinct from the typical CSF and does not seem physiologically plausible. Some comment would be helpful to interpret this result.

We apologise for the confusion caused in the original presentation of this figure. In that figure we set sensitivity at '0' for those frequencies at which subjects did not perform better than chance at our highest available contrast. We were unable to sample contrasts higher than ~18% Michelson contrast without introducing cone contrast. It is entirely possible that we would see detection at lower spatial frequencies if we were able to employ higher contrasts, giving a more standard tail to this response. We appreciate that our original depiction of these data was misleading in implying that there was no sensitivity at higher frequencies and so have removed this panel from figure 2.

2) Fig. 2f: The authors state "Subjects were unable to reliably report the orientation of metameric gratings at any of the spatial frequencies tested..." However, subjects achieved 50% correct performance at 1 cpd. This seems to meet the definition of "reliable." Is the non-statically significant result due to low power (small N)?

We agree that this is possible (and indeed would be consistent with the bandpass tuning of rod-based vision). We have changed the wording of the manuscript to make it clear that we did not aim to definitively determine whether these stimuli are visible at

the dim background, but rather whether there was a qualitative difference between subject performance at this dim vs the brighter irradiance.

3) Fig. 2h: For unlimited exposure duration, subjects achieved approximately 90% correct performance for the melanopic stimuli (Fig. 2a), whereas they only achieved approximately 70% correct performance for an 8 sec exposure duration (Fig. 2h). Note that the function in Fig. 2h asymptotes at 4 sec, which would not predict improvement for longer durations. This discrepancy would be apparent if the authors plotted the unlimited duration data on Fig. 2h (e.g. at minus infinity Hz). Do the authors propose that melanopsin has an integration duration of more than 8 sec? If so, it would be helpful to see measurements for longer exposure durations. For the unlimited duration condition, did the subjects typically view the stimulus for more than 8 sec before making a response?

We agree that one would expect the function at Fig 2H (Fig 2g in updated figures) to asymptote at 90% correct. We interpret its failure to do so as a reflection of the difficulty in maintaining concentration for this task, and the sensitivity of the task to lapses in attention. Participants found detecting and reconstructing the orientation of melanopic gratings demanding and, as many of the stimuli in this task were undetectable (presented at higher temporal frequencies), we asked the participants in this task to maintain high concentration without always having a positive target to detect. In the inverted gratings paradigm the stimulus is expected to be only intermittently visible and there is no external cue to tell the subject when it will be visible. For these reasons, performance in the inverting gratings will have been more sensitive to lapses in attention/concentration than in the stationary gratings experiments. We have changed the wording of the results section to make it clear that we are using this paradigm primarily to probe the limits of melanopsin vision at high temporal frequencies, and also introduced a sentence in the methods describing considerations in interpreting performance at lower temporal frequencies.

REVIEWERS' COMMENTS:

Reviewer #1 (Remarks to the Author):

In my view the authors have dealt clearly and comprehensively with the points raised by both reviewers. Changes to the manuscript are clearly shown and the revision to my mind is perfectly acceptable.

Reviewer #2 (Remarks to the Author):

The authors have responded thoughtfully to my concerns. The manuscript is improved and I have no further suggestions. I congratulate the authors on a nice contribution to the literature.

Jason McAnany
Chicago, IL